# Serrated polyps in patients with ulcerative colitis: Unique clinicopathological and biological characteristics

**Masafumi Nishio[1], Reiko Kunisaki[1]\*, Wataru Shibata[1], Yoichi Ajioka[2], Kingo Hirasawa[3‡], Akiko Takase[4‡], Sawako Chiba[4], Yoshiaki Inayama[4‡], Wataru Ueda[5‡], Kiyotaka Okawa[5‡], Haruka Otake[1‡], Tsuyoshi Ogashiwa[1‡], Hiroto Kinoshita[1‡], Yusuke Saigusa[6‡], Hideaki Kimura[1‡], Jun Kato[7], Shin Maeda[8]**

1 Inflammatory Bowel Disease Center, Yokohama City University Medical Center, Yokohama, Japan,
2 Division of Molecular and Diagnostic Pathology, Niigata University Graduate School of Medical and Dental Sciences, Niigata, Japan, 3 Division of Endoscopy, Yokohama City University Medical Center, Yokohama, Japan, 4 Division of Pathological Diagnosis, Yokohama City University Medical Center, Yokohama, Japan, 5 Department of Gastroenterology and Hepatology, Osaka City Juso Hospital, Osaka, Japan, 6 Department of Biostatistics, Yokohama City University Graduate School of Medicine, Yokohama, Japan, 7 Department of Gastroenterology, Graduate School of Medicine, Chiba University, Chiba, Japan, 8 Department of Gastroenterology, Yokohama City University Graduate School of Medicine, Yokohama, Japan

☯ These authors contributed equally to this work.
‡ KH, AT, YI, WU, KO, HO, TO, HK, YS and HK also contributed equally to this work.
\* reikok@yokohama-cu.ac.jp

**Data Availability Statement:** All relevant data are within the paper and its Supporting information files.

## Abstract

### Background

Serrated polyps have recently been reported in patients with ulcerative colitis (UC); however, their prevalence and detailed characteristics remain unclear.

### Methods

The prevalence and clinicopathological and biological characteristics of serrated polyps in patients with UC were retrospectively examined in a single tertiary inflammatory bowel disease center in Japan from 2000 to 2020.

### Results

Among 2035 patients with UC who underwent total colonoscopy, 252 neoplasms, including 36 serrated polyps (26 in colitis-affected segments, 10 in colitis-unaffected segments), were identified in 187 patients with UC. The proportion of serrated polyps was 1.8% (36/2035). Serrated polyps in colitis-affected segments were common with extensive colitis (88%), history of persistent active colitis (58%), and long UC duration (12.1 years). Serrated polyps in colitis-affected segments were more common in men (88%). Of the 26 serrated polyps in colitis-affected segments, 15, 6, and 5 were categorized as sessile serrated lesion-like dysplasia, traditional serrated adenoma-like dysplasia, and serrated dysplasia not otherwise specified, respectively. Sessile serrated lesion-like dysplasia was common in the proximal colon (67%) and with *BRAF* mutation (62%), whereas traditional serrated adenoma-like

**Funding:** The author(s) received no specific funding for this work.

**Competing interests:** The authors have declared that no competing interests exist.

dysplasia and serrated dysplasia not otherwise specified were common in the distal colon (100% and 80%, respectively) and with *KRAS* mutations (100% and 75%, respectively).

## Conclusions

Serrated polyps comprised 14% of the neoplasias in patients with UC. Serrated polyps in colitis-affected segments were common in men with extensive and longstanding colitis, suggesting chronic inflammation in the development of serrated polyps in patients with UC.

## Introduction

Recent investigations into the pathogenesis of colorectal cancer (CRC) in the general population have indicated that 15% to 35% of CRCs arise through the serrated neoplasia pathway, which differs from the adenoma–carcinoma sequence [1–5]. Among serrated polyps, sessile serrated lesions (SSLs) and traditional serrated adenomas (TSAs) are considered premalignant lesions. In the general population, the prevalences of SSLs and TSAs are 5%–15% and <1%, respectively [6]. *BRAF* mutations, CpG island hypermethylation phenotype (CIMP)-positive status, and microsatellite instability are associated with SSL development, whereas *KRAS* mutations are involved in TSA development.

Ulcerative colitis (UC) is associated with an increased risk of developing CRC [7, 8]. CRC in patients with UC develops through a carcinogenesis pathway distinct from the dominant pathway in sporadic CRC [9]. Risk factors for CRC include a long disease duration, extensive colitis, and more severe or persistent inflammation [7, 8].

Serrated polyps in patients with UC have been reported in several recent articles [10–22]. In these studies, serrated polyps were found in 1.2%–1.7% of patients with UC [16, 20] and accounted for 11%–23% of the neoplasias in patients with UC [20, 23]. Serrated epithelial change has been described as a potential risk factor and precursor of colorectal dysplasia and cancer in patients with UC [22, 24–26]. Serrated polyps in patients with UC have been recognized as histologically and biologically distinct from serrated epithelial change, which shows a high frequency of *TP53* mutations and low frequency of *KRAS/BRAF* mutations [26]. Ko et al. [16] examined serrated polyps in 78 patients with inflammatory bowel disease (IBD), including 56 with UC. The authors reported that SSLs occurred mainly in the proximal colon and contained the *BRAF* mutation, whereas TSAs occurred mainly in the distal colon of men and contained *KRAS* mutations. Miller et al. [21] examined TSA-like lesions in 30 patients with IBD, including 22 with UC, and reported that *KRAS* and *BRAF* mutations were detected in 59% and 16% of the patients, respectively. With the increasing number of reports of serrated polyps in patients with UC, these polyps have become recognized as dysplastic variants that are histologically distinct from conventional dysplasia [27–29]. The most recent histological studies classified serrated polyps in patients with UC into three subtypes: SSL-like dysplasia, TSA-like dysplasia, and serrated dysplasia not otherwise specified (SD NOS) [27, 28]. However, the histological terminology differed among the previous studies, and few studies have examined the clinical characteristics (including risk factors for development) and biological characteristics (including the *KRAS*, *BRAF*, and CIMP status) of serrated polyps in patients with UC based on such accurate histological classification. Therefore, the present study was performed to clarify the yet unknown prevalence, risk factors, and clinical and biological characteristics of serrated polyps in patients with UC.

## Materials and methods

### Data collection, patients, and neoplasia

Consecutive patients with UC who underwent total colonoscopy at Yokohama City University Medical Center, a tertiary IBD center in Japan, from 2000 to 2020 were identified in this retrospective single-center study. Colonoscopies were performed by experienced endoscopists with knowledge of IBD. In particular, for patients with a >7-year disease duration, surveillance colonoscopies with panchromoendoscopy were performed by one of three experienced endoscopists (MN, TO, or RK). The inclusion criteria for endoscopic resection in UC patients were as follows: neoplasias that were well-circumscribed endoscopically, neoplasias that had no evidence of invisible dysplasia in the surrounding mucosa on the basis of confirmational biopsies, and neoplasias that had no evidence of submucosal invasion. When the lesions were diagnosed as neoplasia on the basis of endoscopic observation or biopsy, endoscopic resection was performed regardless of size. When the neoplasia could not be resected endoscopically, surgical resection was selected.

Established UC databases and endoscopy and pathology reports were reviewed to ensure complete case capture. Patients in whom the diagnosis of UC was uncertain, or who had an IBD unclassified status or underwent only flexible sigmoidoscopy, were excluded. Among the patients with UC who underwent total colonoscopy, data for those with colonic neoplasia were extracted. Indefinite for dysplasia and hyperplastic polyps were not considered neoplastic. Data on the demographic and clinical parameters (sex, age at diagnosis of neoplasia and UC, disease duration, disease extent, course of disease, location and endoscopic features of the neoplasia, and treatment of the neoplasia) were obtained from the medical charts.

To compare the clinical and endoscopic findings between serrated polyps in patients with UC and without IBD, we reviewed 249 consecutive serrated polyps in patients without IBD (216 SSLs and 30 TSAs) that were resected endoscopically or surgically at Yokohama City Medical Center during the study period.

### Classification and histological evaluation of neoplasia

All pathological slides of neoplasia were re-reviewed and classified in accordance with the most recent World Health Organization classification [30] by two expert gastrointestinal pathologists (YA and SC); one of whom (YA) was highly specialized in UC-associated dysplasia or cancer.

We classified neoplasia into five categories on the basis of the histological findings and whether the neoplasia was located in colitis-affected segments, as follows: serrated polyps in colitis-affected segments, conventional dysplasia (intestinal-type dysplasia in colitis-affected segments), serrated polyps in colitis-unaffected segments (usually proximal to the extent of the colitis), sporadic adenomas in colitis-unaffected segments, and invasive carcinoma. Additionally, serrated polyps in colitis-affected segments were histologically classified into three subtypes in accordance with previous studies [27–29], as follows: SSL-like dysplasia, TSA-like dysplasia, and serrated SD NOS. Briefly, SSL-like dysplasia is characterized by distorted serrated crypts with prominent basal crypt dilatation (i.e., dilated L- or inverted T-shaped crypts) at the interface with the muscularis mucosa. TSA-like dysplasia is characterized by a villiform growth pattern with columnar cells with intensely eosinophilic cytoplasm and ectopic crypts, creating a prominent serrated profile. SD NOS was defined as serrated dysplasia without definite features of SSL-like or TSA-like dysplasia, with a complex serrated architecture and evidence of dysplasia [27–29]. The representative endoscopic and histological features of SSL-like dysplasia, TSA-like dysplasia, and SD NOS are shown in S1 Fig. Serrated polyps in colitis-

unaffected segments were classified as hyperplastic polyps, SSLs, TSAs, or unclassified serrated adenomas [30].

### Definitions

Persistent active colitis was defined as endoscopically active colitis lasting >6 months. Whether neoplasia was located in colitis-affected segments was determined on the basis of clinical and endoscopic data for the most active inflammation during the course of UC treatment. The proximal colon comprised the cecum, ascending colon, and transverse colon, whereas the distal colon comprised the descending colon, sigmoid colon, and rectum.

### DNA extraction and genetic and epigenetic analyses

DNA samples were purified from archived formalin-fixed, paraffin-embedded blocks of serrated polyps and invasive cancer that had been endoscopically or surgically resected from patients with UC. The pathologists selected the appropriate tissue blocks for DNA extraction. We used a laser microdissection system for selective isolation of the neoplastic sections, avoiding foci of inflammation, and then extracted the DNA. Neoplasia specimens containing adequate DNA quantity and quality were provided for genetic analysis. *KRAS* and *BRAF* mutations were detected using a droplet digital polymerase chain reaction system. The CIMP status was evaluated in accordance with previous reports [31, 32]. The details of the laser microdissection system, DNA extraction, and genetic and epigenetic analyses are shown in S1 File.

### Statistical analysis

Data were analyzed using JMP Pro 12 (SAS Institute Inc., Cary, NC). The prevalence and clinical characteristics were compared among neoplasia groups using Fisher's exact test or the Wilcoxon rank sum test. Statistical significance was set at $P < 0.05$.

### Ethical considerations

This study was approved by the Ethics Committee of Yokohama City University Medical Center (Protocol number: A130926011). All patients with UC whose neoplastic DNA was extracted and genetically analyzed provided written informed consent in accordance with the tenets of the Declaration of Helsinki. An opt-out for the present study was published on the Web.

## Results

### Prevalence of serrated polyps in patients with UC

Fig 1 shows an overview of the neoplasias among the consecutive patients with UC included in the present study. During the study period, 2035 patients with UC underwent total colonoscopy, and 252 neoplasms from 187 patients were identified. Of 219 neoplasms (after excluding 33 invasive cancers), 26 serrated polyps and 132 cases of conventional dysplasia were observed in colitis-affected segments, whereas 10 serrated polyps and 51 sporadic adenomas were found in colitis-unaffected segments.

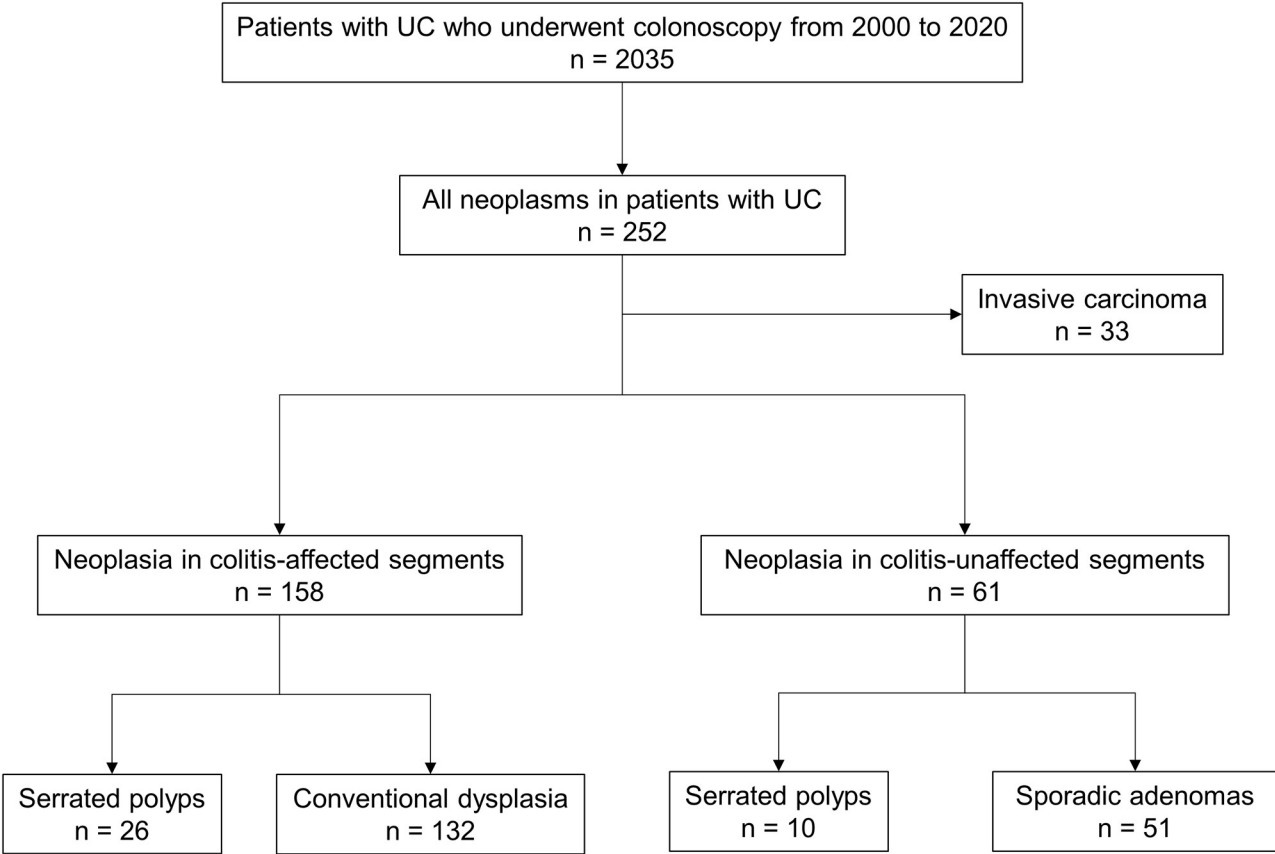

**Fig 1. Overview of the cohort in the present study.** In total, 252 neoplasms were detected in 2035 patients with UC who underwent total colonoscopy during the study period: 158 neoplasms in colitis-affected segments (26 serrated polyps and 132 cases of conventional dysplasia), 61 neoplasms in colitis-unaffected segments (10 serrated polyps and 51 sporadic adenomas), and 33 invasive carcinomas. UC, ulcerative colitis.

## Clinical characteristics of patients with serrated polyps and other neoplasms

Table 1 shows the endoscopic characteristic of serrated polyps in patients with UC and without IBD. Serrated polyps in patients with UC were more frequent in the distal colon and significantly smaller in size compared with those in patients without IBD (42% vs. 24%, respectively, $P = 0.04$, and 10 mm vs. 16 mm, respectively, $P < 0.01$). However, there was no significant defference in neoplasia morphology and border description. Regarding the subtype of serrated polyps (SSL/SSL-like dysplasia and TSA/TSA-like dysplasia), there were no significant differences in neoplasia location, neoplasia morphology, and border description (S1 and S2 Tables).

Table 2 shows the patient and neoplasia characteristics of serrated polyps in colitis-affected and -unaffected segments. Fig 2 shows representative results of the comparisons among the categories of neoplasia (serrated polyps in colitis-affected and colitis-unaffected segments, conventional dysplasia in colitis-affected segments, and sporadic adenomas). The detailed characteristics of each category are shown in S3 Table. The age at diagnosis of neoplasia, age at diagnosis of UC, and duration of UC did not differ between patients with serrated polyps and those with conventional dysplasia found in colitis-affected segments (50 vs. 57 years, $P = 0.11$; 37 vs. 40 years, $P = 0.055$; and 12.1 vs. 10.0 years, $P = 0.21$, respectively) (Fig 2A–2C). However,

**Table 1. Comparison of the endoscopic findings for serrated polyps between patients with UC vs. without IBD.**

|  | Serrated polyps in patients with UC (n = 36) | Serrated polyps in patients without IBD (n = 249) |
|---|---|---|
| Men, n (%) | 25 (69) | 130 (52) |
| Age at diagnosis of neoplasia, years (range) [a] | 49 (39–78) | 64 (52–70) |
| Neoplasia location, n (%) [a] |  |  |
| Proximal colon | 21 (58) | 188 (76) |
| Distal colon | 15 (42) | 61 (24) |
| Size, mm [a] | 10 (7–18) | 16 (10–22) |
| Morphology, n (%) |  |  |
| Polypoid | 6 (17) | 35 (14) |
| Non-polypoid | 30 (83) | 214 (86) |
| Distinct border, n (%) | 36 (100) | 249 (100) |

IBD, inflammatory bowel disease; UC, ulcerative colitis

[a] $P < 0.05$ between serrated polyps in patients with UC and without IBD

the age at diagnosis of neoplasia and age at diagnosis of UC in patients with sporadic adenomas (67 and 52 years, respectively) were significantly older than those in patients with each neoplasia in colitis-affected segments (*P* values are shown in Fig 2A and 2B). Serrated polyps in colitis-affected segments were more frequent in men than in women, while those in colitis-unaffected segments were more frequent in women than in men (percentages in men: 88% vs. 20%, respectively; *P* < 0.001) (Fig 2D).

Patients with either type of neoplasia in colitis-affected segments were more likely to have extensive colitis (serrated polyps: 88%, conventional dysplasia: 74%) and a history of persistent active colitis (serrated polyps: 58%, conventional dysplasia: 52%) compared with patients with either type of neoplasia in colitis-unaffected segments (serrated polyps: 40% and 0%, respectively; sporadic adenomas: 10% and 22%, respectively) (*P* values are shown in Fig 2E and 2F). Neoplasia size was similar (serrated polyps in colitis-affected segments: 9 [interquartile range: 7–19] mm, conventional dysplasia: 9 [6–18] mm, serrated polyps in colitis-unaffected segments: 10 [8–19] mm), with the exception of sporadic adenomas (4 [3–5] mm) (*P* values are shown in Fig 2G). Regarding previous treatment, immunomodulators were used significantly more frequently in patients with serrated polyps in colitis-affected segments than in those with polyps in colitis-unaffected segments.

## Clinical characteristics of serrated polyps in colitis-affected segments

Of 26 serrated polyps in colitis-affected segments, 15 (58%), 6 (23%), and 5 (19%) were categorized as SSL-like dysplasia, TSA-like dysplasia, and SD NOS, respectively. The characteristics of these polyps are shown in Table 3. All subtypes of serrated polyps in colitis-affected segments were common in men. SSL-like dysplasia was common in the proximal colon, whereas TSA-like dysplasia and SD NOS were common in the distal colon. Although statistical analysis was not performed owing to the small number of samples, SD NOS was more likely to be associated with large neoplasia size (23 mm) than with SSL-like dysplasia (12 mm) or TSA-like dysplasia (8 mm). TSA-like dysplasia was more likely to have a polypoid morphology (67%) compared with SSL-like dysplasia (7%) and SD NOS (20%).

**Table 2. Clinical characteristics of serrated polyps in patients with UC.**

| | Serrated polyps in patients with UC | |
|---|---|---|
| | In colitis-affected segments (n = 26) | In colitis-unaffected segments (n = 10) |
| Men [a] | 23 (88) | 2 (20) |
| Age at diagnosis of neoplasia, years | 50 (39–62) | 44 (38–78) |
| Age at diagnosis of UC, years | 37 (22–43) | 39 (34–58) |
| Duration of UC at diagnosis of neoplasia, years [a] | 12.1 (7.8–20.4) | 5.5 (1.2–17.1) |
| Disease type | | |
| Extensive colitis (E3) [a] | 23 (88) | 4 (40) |
| Left-sided colitis (E2) | 2 (8) | 2 (20) |
| Proctitis (E1) [a] | 1 (4) | 4 (40) |
| History of persistent active colitis | 15 (58) | 2 (20) |
| History of severe disease in UC | 6 (23) | 0 (0) |
| Previous treatment, n (%) | | |
| 5-ASA/SASP | 25 (96) | 8 (80) |
| Corticosteroid | 17 (62) | 2 (20) |
| Immunomodulator | 12 (47) | 2 (20) |
| Anti-TNF antibody | 6 (23) | 1 (10) |
| Calcineurin inhibitor | 3 (12) | 0 (0) |
| JAK inhibitor | 0 (0) | 0 (0) |
| $\alpha 4\beta 7$ inhibitor | 0 (0) | 0 (0) |
| IL12/23 inhibitor | 0 (0) | 0 (0) |
| Neoplasia location [a] | | |
| Proximal colon | 11 (42) | 10 (100) |
| Distal colon | 15 (58) | 0 (0) |
| Size, mm | 9 (7–19) | 10 (8–19) |
| Morphology | | |
| Polypoid | 6 (23) | 2 (20) |
| Non-polypoid | 20 (77) | 8 (80) |

Data are presented as n (%) or median (interquartile range).

IL, interleukin; JAK, Janus kinase; SASP, salazosulfapyridine; TNF, tumor necrosis factor; UC, ulcerative colitis; 5-ASA, 5-aminosalicylic acid

[a] $P < 0.05$ between serrated polyps in colitis-affected segments and serrated polyps in colitis-unaffected segments

## Locational and biological characteristics of serrated polyps in patients with UC

Fig 3 shows the locational distribution of serrated polyps in patients with UC, including 26 polyps in colitis-affected segments (Fig 3A) and 10 polyps (9 SSLs and 1 TSA) in colitis-unaffected segments (Fig 3B). SSL-like dysplasia was distributed in all segments of the colorectum, and 10 lesions (67%) were located in the proximal colon. All cases of TSA-like dysplasia were located in the distal colon (2 [33%] in the rectum and 4 [67%] in the sigmoid colon). Four (80%) SD NOS lesions were located in the distal colon (one in the descending colon and three in the rectum), and one was located in the proximal colon. In contrast, all serrated polyps in colitis-unaffected segments were located in the proximal colon.

Of 36 serrated polyps in both colitis-affected and -unaffected segments in patients with UC, genetic and epigenetic evaluations were performed in 18 specimens (13 affected and 5

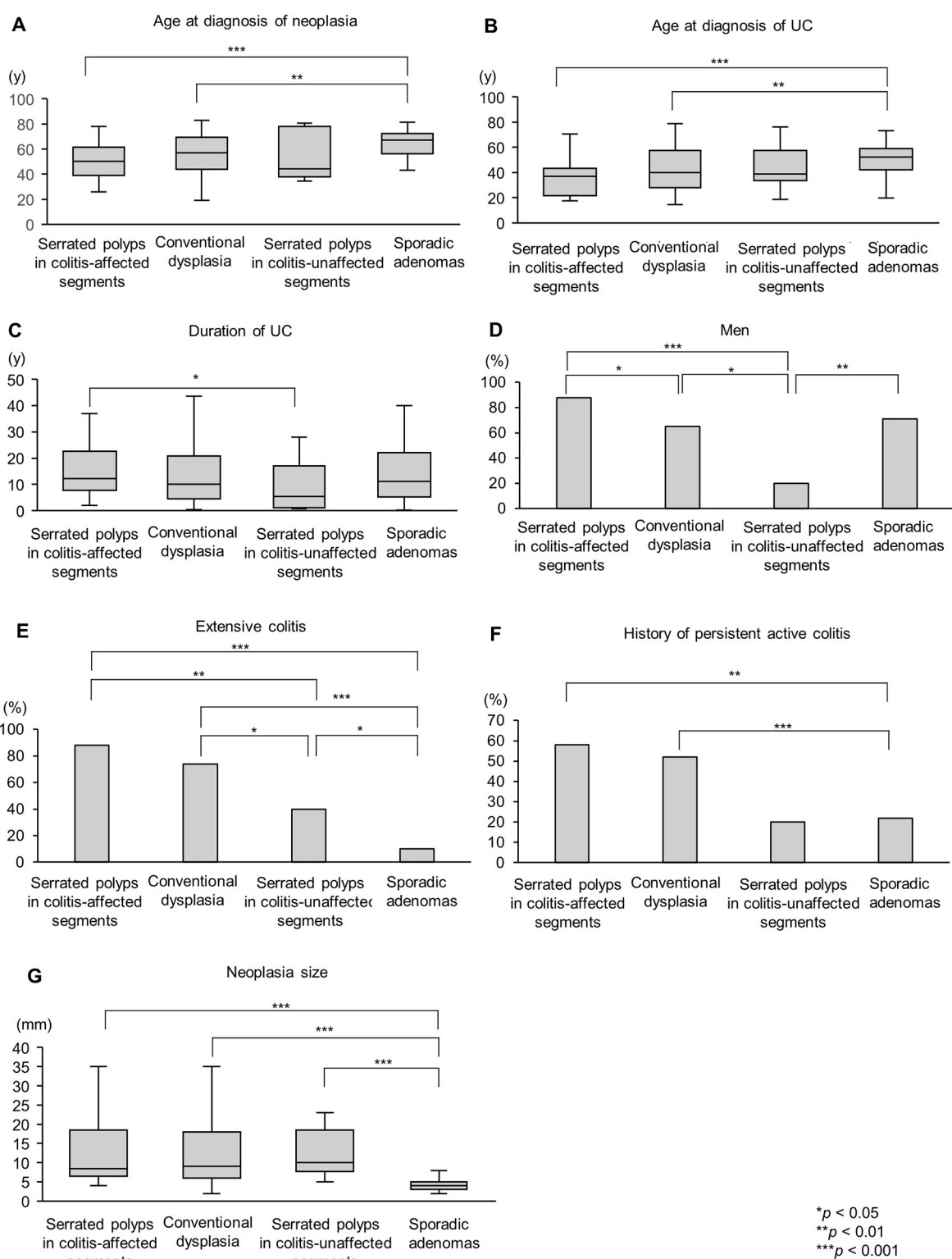

**Fig 2. Comparison of the major clinical characteristics among the neoplasia categories.** (A) age at diagnosis of neoplasia, (B) age at diagnosis of UC, (C) duration of UC, (D) ratio of men, (E) proportion of extensive colitis, (F) proportion of patients with a history of persistent active colitis, and (G) neoplasia size. Fisher's exact test and the Wilcoxon rank sum test were used for the statistical analysis. Statistical significance was set at $P < 0.05$. *$P < 0.05$, **$P < 0.01$, and ***$P < 0.001$. UC, ulcerative colitis.

**Table 3. Clinical characteristics of each subtype of serrated polyp in colitis-affected segments.**

| | SSL-like dysplasia (n = 15) | TSA-like dysplasia (n = 6) | SD NOS (n = 5) |
|---|---|---|---|
| Men | 13 (87) | 6 (100) | 4 (80) |
| Age at diagnosis of neoplasia, years | 49 (40–62) | 44 (26–69) | 52 (48–62) |
| Age at diagnosis of UC, years | 39 (23–44) | 30 (19–60) | 36 (23–42) |
| Duration of UC at diagnosis of neoplasia, years | 13.7 (9.2–22.5) | 7.8 (6.3–13.5) | 25.5 (5.3–34.5) |
| Disease type | | | |
| Extensive colitis (E3) | 13 (86) | 6 (100) | 4 (80) |
| Left-sided colitis (E2) | 1 (7) | 0 (0) | 1 (20) |
| Proctitis (E1) | 1 (7) | 0 (0) | 0 (0) |
| History of persistent active colitis | 8 (53) | 4 (67) | 3 (60) |
| History of severe disease | 1 (7) | 4 (67) | 1 (20) |
| Previous treatment, n (%) | | | |
| 5-ASA/SASP | 14 (93) | 6 (100) | 5 (100) |
| Corticosteroid | 9 (60) | 4 (67) | 4 (80) |
| Immunomodulator | 6 (40) | 4 (67) | 2 (40) |
| Anti-TNF antibody | 4 (27) | 0 (0) | 2 (40) |
| Calcineurin inhibitor | 1 (7) | 1 (17) | 1 (20) |
| JAK inhibitor | 0 (0) | 0 (0) | 0 (0) |
| α4β7 inhibitor | 0 (0) | 0 (0) | 0 (0) |
| IL12/23 inhibitor | 0 (0) | 0 (0) | 0 (0) |
| Neoplasia location | | | |
| Proximal colon | 10 (67) | 0 (0) | 1 (20) |
| Distal colon | 5 (33) | 6 (100) | 4 (80) |
| Size, mm | 8 (7–15) | 7 (5–13) | 20 (19–29) |
| Morphology | | | |
| Polypoid | 1 (7) | 4 (67) | 1 (20) |
| Non-polypoid | 14 (93) | 2 (33) | 4 (80) |

Data are presented as n (%) or median (interquartile range).

IL, interleukin; JAK, Janus kinase; SASP, salazosulfapyridine; SD NOS, serrated dysplasia not otherwise specified; SSL, sessile serrated lesion; TNF, tumor necrosis factor; TSA, traditional serrated adenoma; UC, ulcerative colitis; 5-ASA, 5-aminosalicylic acid

unaffected) from which DNA of adequate quality and quantity was obtained (Table 4). *BRAF* mutations were observed in 75% (3/4) of cases of SSL-like dysplasia in colitis-affected segments and in all SSLs in colitis-unaffected segments, whereas *KRAS* mutations were observed in all cases of TSA-like dysplasia in colitis-affected segments and in all TSAs in colitis-unaffected segments. In SD NOS, 75% (3/4) of the specimens showed *KRAS* mutations and 25% (1/4) showed *BRAF* mutations. CIMP-positive status for serrated polyps in colitis-affected segments was observed in 60% (3/5) of cases of TSA-like dysplasia but in only 25% of cases of SSL-like dysplasia and SD NOS. The genetic and epigenetic status of 16 invasive cancers in colitis-affected segments were also evaluated. *KRAS* and *BRAF* mutations and CIMP-positive status were observed in 13% (2/16), 0% (0/16), and 19% (3/16) of invasive cancers in colitis-affected segments (S4 Table).

## Discussion

This study showed the prevalence of serrated polyps in patients with UC and is the first, to our knowledge, to suggest the increasing prevalence of these polyps, which have been observed

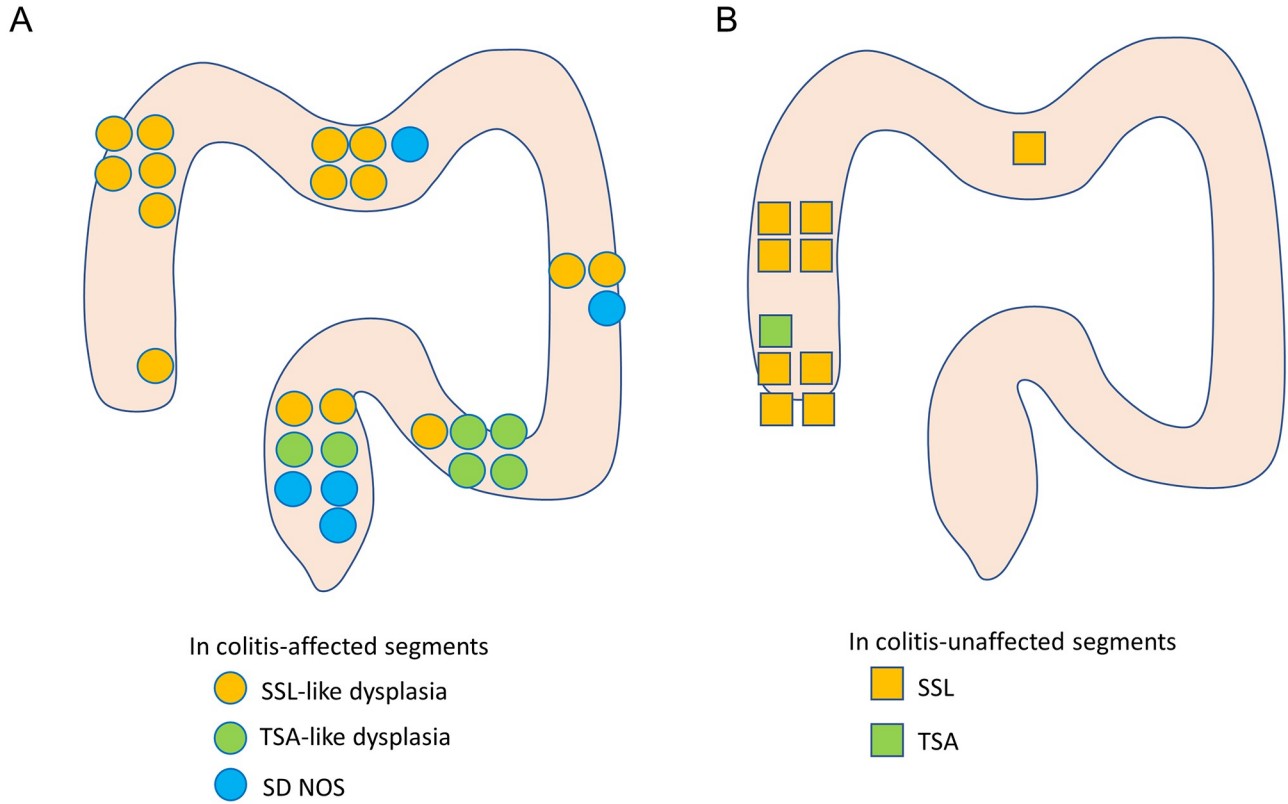

**Fig 3. Distribution of serrated polyps in patients with UC.** (A) Among serrated polyps in colitis-affected segments, 15 lesions were SSL-like dysplasia, 6 were TSA-like dysplasia, and 5 were SD NOS. (B) Among serrated polyps in colitis-unaffected segments, nine lesions were SSLs and one was a TSA. SD NOS, serrated dysplasia not otherwise specified; SSL, sessile serrated lesion; TSA, traditional serrated adenoma.

more frequently recently. The clinical and biological characteristics of SSL-like dysplasia and TSA-like dysplasia in patients with UC are basically similar to those of each counterpart in individuals without IBD. However, SSL-like dysplasia in patients with UC was more frequently observed in men than in women, in the present study, a finding that contrasts the known female dominance of SSLs in individuals without IBD. Although many other issues remain unknown, our findings suggest a new pathway in the development of UC-associated neoplasia.

The present study showed that the prevalence of serrated polyps was 1.8% (36/2035) in patients with UC, and these polyps accounted for 14% (36/252) of all neoplasias in patients with UC. This proportion is similar to that in a previous study that reported a prevalence of

**Table 4. Genetic and epigenetic analyses of serrated polyps in patients with ulcerative colitis.**

| | In colitis-affected segments | | | | In colitis-unaffected segments | | |
|---|---|---|---|---|---|---|---|
| | Total (n = 13) | SSL-like dysplasia (n = 4) | TSA-like dysplasia (n = 5) | SD NOS (n = 4) | Total (n = 5) | SSL (n = 4) | TSA (n = 1) |
| *KRAS* | 8 (62) | 0 (0) | 5 (100) | 3 (75) | 1 (20) | 0 (0) | 1 (100) |
| *BRAF* | 4 (31) | 3 (75) | 0 (0) | 1 (25) | 4 (80) | 4 (100) | 0 (0) |
| CIMP-positive | 5 (38) | 1 (25) | 3 (60) | 1 (25) | 4 (80) | 3 (75) | 1 (100) |

Data are presented as n (%).

CIMP, CpG island methylation phenotype; SD NOS, serrated dysplasia not otherwise specified; SSL, sessile serrated lesion; TSA, traditional serrated adenoma

serrated polyps of 1.2%–1.7% in patients with IBD [16, 20]. Additionally, serrated polyps accounted for 11%–23% of all neoplasias in the study [20, 23].

Regarding ethnicity, the present study is the first, to our knowledge, to investigate the prevalence of serrated polyps in the Asian population. Meta-analyses of the non-IBD population have reported that the prevalence of serrated polyps in Western countries was higher than those in Eastern countries [33, 34]. The possible reason for the similar proportions of serrated polyps in patients with UC between the present study and previous studies is differences in the study periods. Two previous studies included serrated polyps detected between 2005 and 2007 [20] and 2000 and 2013 [16]. Because serrated polyps (SSLs and TSAs) were considered hyperplastic polyps before 2010, SSLs and TSAs may have been overlooked without biopsy or endoscopic resection. To clarify the differences in the prevalence between Eastern and Western countries, an accumulation of recent cases in both ethnic groups is desirable.

Previous studies have shown that serrated polyps in patients with UC have clinical and biological characteristics similar to those in individuals without IBD. For instance, SSLs are usually found in women and in the proximal colon, whereas TSAs are usually found in men and in the distal colon [16, 21]. Our results also showed similar neoplasia location, neoplasia morphology, and border description for both SSLs/SSL-like dysplasias and TSAs/TSA-like-dysplasias in patients with UC and without IBD. The difference in the neoplasia size between the groups may be associated with different indications for endoscopic resection between patients with UC and without IBD. Notably, our results demonstrated a unique clinical characteristic in SSL-like dysplasia in colitis-affected segments. SSLs in colitis-unaffected segments showed clinical and biological similarity to those in patients without IBD (all SSLs in colitis-unaffected segments were found in women and in the proximal colon). In contrast, most cases of SSL-like dysplasia in colitis-affected segments were found in men, although their locational and biological characteristics were similar to those of SSLs in colitis-unaffected segments (predominantly in the proximal colon and with *BRAF* mutation). These sex-related differences may arise from differences in the biological mechanisms of development between serrated pathways with and without background inflammation. In this context, although previous studies indicated that SSLs in patients with UC were dominant in women and in the proximal colon, the distribution of SSLs was not always based on the presence or absence of background inflammation [17, 18, 35]. Conversely, TSA-like dysplasia in colitis-affected segments showed clinical and biological characteristics similar to those of TSAs in colitis-unaffected segments (predominantly in men, in the distal colon, and with *KRAS* mutations). These results suggest that the specific development of SSL-like dysplasia in men with UC is derived from a unique tumorigenic pathway.

Serrated polyps in colitis-affected segments were more common in patients with a long disease duration and a history of persistent active colitis than in patients without these characteristics, and this was also true in the development of UC-related dysplasia. Therefore, chronic inflammation must be involved in the development of both dysplasia and serrated polyps in patients with UC. Notably, both patients who have IBD with dysplasia and patients without IBD who have serrated polyps are likely to develop synchronous and metachronous CRCs. Considering this similarity, serrated polyps in patients without IBD may also develop on a background of chronic inflammation. In fact, previous studies have shown that the concentrations of some key inflammatory factors, such as tumor necrosis factor α, cyclooxygenase-2, interleukin-4, and interleukin-1β, are increased in patients with serrated polyps without IBD [36, 37]. These studies suggest that chronic inflammation is correlated with the development of serrated polyps in both patients with UC and individuals without IBD. Long-term follow-up of our patients and other observational studies are needed to further examine the multiple developmental pathways of serrated polyps in patients with UC.

Recently, SD NOS was newly described as serrated polyps without definite features of SSL-like and TSA-like dysplasia [28]. However, only one study [28] reported a small number of cases of SD NOS, and the clinical and biological characteristics were not assessed. The present study is the first, to our knowledge, to show the clinical and biological characteristics of SD NOS. Our study showed a similar distribution (predominantly located in the distal colon) and similar biological characteristics between TSA-like dysplasia and SD NOS, suggesting that SD NOS is a subtype of TSA-like dysplasia. Additionally, considering the longer duration of UC and larger size of SD NOS compared with the duration and size of TSA-like dysplasia, SD NOS may be an enlarged lesion of TSA-like dysplasia induced by long-term inflammation. Similarly, Ko et al. [16] reported that 32% of the serrated polyps in patients with IBD were TSA-like, and this rate is consistent with our rate of the combination of TSA-like dysplasia and SD NOS (42%). Although the reason for the higher proportion of TSA-like dysplasia (with SD NOS) than the proportion of TSAs in patients without IBD is unclear, chronic inflammation may also be involved in the development of TSA-like dysplasia.

No previous studies have assessed the CIMP status of serrated polyps in patients with UC. Although CIMP-positive status is important in the serrated neoplasia pathway, recent studies showed that CIMP-positive status was exclusively associated with SSLs in the proximal colon and advanced age [38]. The low rate of a CIMP-positive status of SSL-like dysplasia in our study was considered to be associated with the neoplasia location (33% were in the distal colon) and relatively young age of our study population. In patients without IBD, TSAs in the proximal colon frequently show *BRAF* mutation and CIMP positivity, whereas TSAs in the distal colon show *KRAS* mutations and CIMP negativity [39, 40]. In our study, although all cases of TSA-like dysplasia were located in the distal colon and showed *KRAS* mutations, 60% (3/5) were also CIMP-positive. The reason for this discrepancy is unclear, but TSA-like dysplasia arising from chronic inflammatory mucosa may develop through a pathway distinct from TSAs in non-inflamed mucosa. The proportion of *KRAS* and *BRAF* mutations and CIMP-positive status in serrated polyps in colitis-affected segments tended to be higher in invasive cancer in colitis-affected segments in this study. These results suggest that most serrated polyps may not be precursors of colitis-associated invasive cancers. Although the optimal management of serrated polyps in patients with UC is still unclear, our results suggest that colectomy is excessive for serrated polyps in colitis-affected segments. However, considering that serrated polyps are known precursors of CRCs in patients without IBD [1–5], serrated polyps in patients with UC should be treated endoscopically as for those in patients without IBD. Although there was no UC-associated dysplasia within SSLs or TSAs in the present study, high-grade dysplasia with a hyperplastic polyp was detected in a 19-year-old man in our hospital (this case was not included in the present study because the lesion did not meet the diagnostic criteria for SSL, TSA, or SD NOS). This case suggests that some serrated polyps in patients with UC, although rare, have malignant potential. To determine the appropriate management of serrated polyps in patients with UC, further accumulation of cases is desirable.

Our study has some limitations. The setting was a single center, and the study had a small sample size, which included only Japanese patients. Undoubtedly, the number of serrated polyps in patients with UC was smaller than those of conventional dysplasia and sporadic adenomas. Another limitation is the lack of data regarding a family histology of CRC, obesity, smoking, and alcohol intake, which are risk factors for SSLs.

## Conclusions

The present study showed the detailed clinicopathological and biological characteristics of serrated polyps in patients with UC. Serrated polyps in colitis-affected segments were common in

men with extensive colitis and a long duration of UC, suggesting that chronic inflammation might be involved in the development of serrated polyps in patients with UC.

## Supporting information

**S1 Fig. Endoscopic and histological views of representative cases of each subtype of serrated polyp in colitis-affected segments.** (A–C) Representative case of SSL-like dysplasia. (A) A 12-mm non-polypoid (superficial elevated) lesion was detected in the sigmoid colon. (B, C) Histologically, distorted serrated crypts with dilated L- and inverted T-shaped crypts are visible, resembling SSLs in individuals without IBD. (C) Magnified view of the box outlined in orange in (B). (D–F) Representative case of TSA-like dysplasia. (D) A 15-mm polypoid (sessile) lesion was detected in the sigmoid colon. (E, F) Histologically, a villous pattern with eosinophilic cytoplasm and ectopic crypt formation is visible, resembling TSAs in individuals without IBD. (F) Microscopic view of the box outlined in orange in (E). (G–I) Representative case of SD NOS. (G) A 20-mm non-polypoid (superficial elevated) lesion was detected in the descending colon. (H, I) Histologically, the lesion has a serrated structure resembling a TSA, but does not show a villous or papillary structure or ectopic crypt formation, which that would have led to a definitive diagnosis of TSA. (I) Microscopic view of box outlined in orange in (H). IBD, inflammatory bowel disease; SD NOS, serrated dysplasia not otherwise specified; SSL, sessile serrated lesion; TSA, traditional serrated adenoma.
(TIF)

**S1 Table. Comparison of the clinical and endoscopic characteristics between SSL/SSL-dysplasias in patients with UC and SSLs in patients without IBD.**
(DOCX)

**S2 Table. Comparison of the clinical and endoscopic characteristics between TSA/TSA-dysplasias in patients with UC and TSAs in patients without IBD.**
(DOCX)

**S3 Table. Comparison of the clinical characteristics among the neoplasia categories.**
(DOCX)

**S4 Table. Genetic and epigenetic analyses of serrated polyps and colitis-associated cancer.**
(DOCX)

**S1 File. Details of the laser microdissection system, DNA extraction, and genetic and epigenetic analyses.**
(DOCX)

**S1 Checklist. STROBE statement—Checklist of items that should be included in reports of observational studies.**
(DOCX)

## Author Contributions

**Conceptualization:** Masafumi Nishio, Reiko Kunisaki, Wataru Shibata, Yoichi Ajioka, Jun Kato, Shin Maeda.

**Data curation:** Masafumi Nishio, Wataru Shibata.

**Formal analysis:** Masafumi Nishio, Yusuke Saigusa.

**Investigation:** Masafumi Nishio, Reiko Kunisaki, Wataru Shibata, Yoichi Ajioka, Sawako Chiba, Yoshiaki Inayama.

**Project administration:** Reiko Kunisaki, Kingo Hirasawa, Jun Kato, Shin Maeda.

**Supervision:** Jun Kato, Shin Maeda.

**Writing – original draft:** Masafumi Nishio, Reiko Kunisaki.

**Writing – review & editing:** Wataru Shibata, Yoichi Ajioka, Kingo Hirasawa, Akiko Takase, Sawako Chiba, Yoshiaki Inayama, Wataru Ueda, Kiyotaka Okawa, Haruka Otake, Tsuyoshi Ogashiwa, Hiroto Kinoshita, Yusuke Saigusa, Hideaki Kimura, Jun Kato, Shin Maeda.

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
