## [Decision Letter · Decision Letter 0]

3 Jan 2023

PONE-D-22-31131Serrated Polyps in Patients with Ulcerative Colitis: Prevalence and Unique Clinicopathological and Biological CharacteristicsPLOS ONE

Dear Dr. Kunisaki,

Thank you for submitting your manuscript to PLOS ONE. After careful consideration, we feel that it has merit but does not fully meet PLOS ONE’s publication criteria as it currently stands. Therefore, we invite you to submit a revised version of the manuscript that addresses the points raised during the review process.

This manuscript was carefully reviewed by 2 experts. Although both reviewers highly evaluated this study, they suggested several points which need to be addressed before acceptance. For instance, it is important to see whether endoscopic findings of serrated lesions in UC-patients differ from those in non-UC patients. Reviewer 1 also suggested additional mutation and methylation analyses, while the editor considers it is not mandatory because KRAS, BRAF mutations and CIMP status are already determined in the samples. Reviewer 2 indicated possible difficulties in comparing samples collected in different time periods because of the inconsistent clinical definition of serrated lesions. Reviewer 2 also suggested additional molecular analysis in in invasive cancer. Analysis of KRAS/BRAF mutations and CIMP status in the invasive cancers may help us to understand whether those cancers develop through the serrated pathway. Please respond to each of the reviewer comments.

We look forward to receiving your revised manuscript.

Kind regards,

Hiromu Suzuki, M.D., Ph.D.

Academic Editor

PLOS ONE

Journal Requirements:

2. You indicated that you had ethical approval for your study. Please clarify whether minors (participants under the age of 18 years) were included in this study. If yes, in your Methods section, please ensure you have also stated whether you obtained consent from parents or guardians of the minors included in the study or whether the research ethics committee or IRB specifically waived the need for their consent.

Reviewers' comments:

Reviewer's Responses to Questions

**Comments to the Author**

1. Is the manuscript technically sound, and do the data support the conclusions?

Reviewer #1: Yes

Reviewer #2: Partly

2. Has the statistical analysis been performed appropriately and rigorously? 

Reviewer #1: Yes

Reviewer #2: Yes

3. Have the authors made all data underlying the findings in their manuscript fully available?

Reviewer #1: Yes

Reviewer #2: Yes

4. Is the manuscript presented in an intelligible fashion and written in standard English?

Reviewer #1: Yes

Reviewer #2: Yes

5. Review Comments to the Author

Reviewer #1: The paper is potentially interesting and seems to be convincing, however, I have several concerns with this paper as shown below.

1. Comparison should be made between the endoscopic findings of serrated lesions in UC patients compared to non-UC patients.

2. The authors should determine mutations of driver genes of colorectal serrated lesions such as APC, GNAS, and RNF43.

3. The authors should analyze all 36 serrated lesions for genetic alterations and methylation statuses.

4. The difference between Asian patients and non-Asian patients is of interest of readers. Are there ethnic differences? The authors are encouraged to add more discussion in details on this point.

5. Grammatical and typographical errors in the text need to be corrected by a native English speaker.

Reviewer #2: In this manuscript, the authors study clinical and molecular features of serrated polyps in patients with ulcerative colitis (UC), retrospectively. Clinicopathological and biological characteristics of serrated polyps in colitis-affected segments is potentially interesting, but several concerns need to be addressed.

1, Neoplasias enrolled in this study was classified into five categories based on histological findings. Were all neoplasias endoscopically or surgically resected? In fig3G, median size of sporadic adenoma was less than 5mm. Criteria of endoscopic resection should be shown.

2, It is difficult to compare the prevalence of serrated polyps in patients with UC from 2000 to 2009 and from 2010 to 2020. Endoscopic findings of serrated polyp were not established, and almost serreted polyps were classified into hyperplastic polyp before 2010, as the authors are discussing. If the authors want to show the increasing incidence of serrated polyps, the data after 2010 should be used.

3, Molecular features of serrated polyp in colitis affected and unaffected segments were interesting. Molecular features of invasive cancer should be examined.

6. PLOS authors have the option to publish the peer review history of their article (what does this mean?). If published, this will include your full peer review and any attached files.

Reviewer #1: No

Reviewer #2: No

---

## [Author Response · Author response to Decision Letter 0]

26 Jan 2023

Responses to the Reviewers’ Comments

Reviewer #1: 

The paper is potentially interesting and seems to be convincing, however, I have several concerns with this paper as shown below.

1. Comparison should be made between the endoscopic findings of serrated lesions in UC patients compared to non-UC patients.

Response:

We thank Reviewer #1 for the insightful comment, which has helped us improve the Results and Discussion sections of our manuscript.

As Reviewer #1 pointed out, a comparison of the clinical and endoscopic characteristics between serrated polyps in patients with UC and without IBD is important. Accordingly, we reviewed 249 consecutive serrated polyps in patients without IBD (216 SSLs and 30 TSAs) that were resected endoscopically or surgically at our hospital during the study period. We compared the endoscopic characteristics of serrated polyps in patients with UC and without IBD. Serrated polyps in patients with UC were more frequent in the distal colon and significantly smaller in size than those in patients without IBD (42% vs. 24%, respectively, P = 0.04, and 10 mm vs. 16 mm, respectively, P < 0.01). However, there was no significant deference in neoplasia morphology and border description. Next, we compared the endoscopic characteristics on the basis of the subtype of serrated polyps (SSL/SSL-like dysplasia and TSA/TSA-like dysplasia). There was no significant deference in neoplasia location, neoplasia morphology, and border description between patients with UC and without IBD for both SSL/SSL-like dysplasia and TSA/TSA-like dysplasia. These results support the finding in previous studies that serrated polyps in patients with UC were similar to those in patients without IBD. However, our results showed a unique characteristic of SSL-like dysplasia in colitis-affected segments: these were common in men. 

We added this information in the methods and discussion of the revised manuscript. We also created new Table 1, S1 Table, and S2 Table. Previous Tables 1, 2, and 3, and S1 Table have been renumbered accordingly.

2. The authors should determine mutations of driver genes of colorectal serrated lesions such as APC, GNAS, and RNF43.

Response:

We thank Reviewer #1 for the insightful comment.

As Reviewer #1 pointed out, APC, RNF43, and GNAS mutations have been reported in addition to KRAS or BRAF mutations in TSAs in patients without IBD. 

Accordingly, we examined somatic mutations in serrated polyps in colitis-affected segments using next-generation sequencing (NGS). However, sufficient quality and quantity of DNA were obtained in only six samples. After excluding samples that did not meet the inclusion criterion of successful sequencing, only three samples remained (two TSA-like dysplasias and one SD NOS). Among the three samples that could be evaluated, KRAS mutations were detected in all cases, whereas APC and GNAS mutations were not detected. 

In our study, RNF43 mutations were not evaluated because the Cancer Hotspot Panel v2 (Thermo Fisher Scientific) that we used did not include RNF43. 

Because only three samples could be examined, the above data were insufficient to discuss somatic mutations other than KRAS and BRAF mutations of serrated polyps in patients with UC. Therefore, we presented only the data for NGS in this response.

3. The authors should analyze all 36 serrated lesions for genetic alterations and methylation statuses.

Response:

We thank Reviewer #1 for the insightful comment. 

As Reviewer #1 pointed out, analysis of all 36 lesions is desirable for more accurate genetic and epigenetic characterization. However, in our study, only 18 samples had sufficient quality or quantity of DNA for analysis. Therefore, it is difficult to perform additional examinations. The possible reasons for not obtaining sufficient quality or quantity of DNA are as follows: First, a long time has passed since the samples were collected. Second, before 2017, the specimens were fixed in 20% formalin buffer, which caused tissue damage.

We would like to examine the DNA of more cases under improved preservation conditions, in the future.

4. The difference between Asian patients and non-Asian patients is of interest of readers. Are there ethnic differences? The authors are encouraged to add more discussion in details on this point.

Response:

We thank Reviewer #1 for the insightful comments, which have helped us improve the Discussion section of our manuscript. 

As Reviewer #1 have pointed out, the prevalence of serrated lesions in the non-IBD population differs between ethnic groups, and it is very important to consider ethnic differences in serrated polyps in patients with UC. 

To the best of our knowledge, no study has compared the prevalence of serrated polyps in patients with UC between Eastern and Western countries. All previous studies that reported the prevalence of serrated lesions in patients with UC were from Western countries. Therefore, to our knowledge, the present study is the first to report the prevalence in the Asian population. 

Two meta-analyses of the non-IBD population reported that the prevalence of serrated polyps in Western countries was higher than those in Eastern countries. In comparison, the prevalence of serrated polyps in patients with UC in our study (1.8%) was similar to those in previous studies from Western countries (1.2%–1.7%). The possible reason for this similarity is differences in the study periods. The two previous studies included serrated polyps detected between 2005 and 2007 and 2000 and 2013, respectively. Because serrated polyps (SSLs and TSAs) were considered hyperplastic polyps before 2010, SSLs and TSAs may have been overlooked without biopsy or endoscopic resection. To clarify the differences in the prevalence between Eastern and Western countries, the accumulation of more recent cases in both ethnic groups is desirable.

We added the above information in the revised discussion.

5. Grammatical and typographical errors in the text need to be corrected by a native English speaker.

Response:

We thank Reviewer #1 for the insightful comment, which has helped us improve our manuscript. 

The revised manuscript has been edited by a native English-speaking Medical Editor.

Reviewer #2: 

In this manuscript, the authors study clinical and molecular features of serrated polyps in patients with ulcerative colitis (UC), retrospectively. Clinicopathological and biological characteristics of serrated polyps in colitis-affected segments is potentially interesting, but several concerns need to be addressed.

1, Neoplasias enrolled in this study was classified into five categories based on histological findings. Were all neoplasias endoscopically or surgically resected? In fig3G, median size of sporadic adenoma was less than 5mm. Criteria of endoscopic resection should be shown.

Response: 

We thank Reviewer #2 for the insightful comments. 

All neoplasias evaluated in the present study were resected endoscopically or surgically. 

As Reviewer #2 pointed out, the Japanese guidelines recommend endoscopic resection for adenomas ≥ 6 mm in size or for superficial depressed-type lesions even when the lesion measures ≤ 5 mm in non-IBD patients. However, no indication for endoscopic resection on the basis of size has been established for colitis-associated dysplasia. In patients with UC, the distinction between sporadic adenoma and colitis-associated dysplasia is sometimes difficult even after observation with magnifying endoscopy and biopsy. Therefore, in our hospital, when the lesions are diagnosed as neoplasia on the basis of endoscopic observation or biopsy, the lesions are resected regardless of size. In this study, the inclusion criteria for endoscopic resection in UC patients were as follows: neoplasias that were well-circumscribed endoscopically, neoplasias that had no evidence of invisible dysplasia in the surrounding mucosa with confirmational biopsies, and neoplasias that had no evidence of submucosal invasion. When the neoplasia could not be resected endoscopically, surgical resection was performed.

We added the criteria for endoscopic resection in patients with UC in the Methods section.

2, It is difficult to compare the prevalence of serrated polyps in patients with UC from 2000 to 2009 and from 2010 to 2020. Endoscopic findings of serrated polyp were not established, and almost serreted polyps were classified into hyperplastic polyp before 2010, as the authors are discussing. If the authors want to show the increasing incidence of serrated polyps, the data after 2010 should be used.

Response:

We thank Reviewer #2 for the insightful comments. 

As Reviewer #2 pointed out, data after 2010 should be used to investigate the increasing incidence of serrated polyps. However, we think that it is inappropriate to discuss the increasing incidence of serrated polyps in post-2010 data on the basis of the present data because of the short study period. Additionally, considering that the definition of serrated polyps was revised in 2019, further accumulation of serrated polys detected after 2019 is needed to investigate the prevalence of serrated polyps.

Therefore, we deleted text that mentioned the increasing incidence of serrated polyps in the abstract, results, and discussion. We also deleted original Fig 2, and current Figs 3 and 4 have been renumbered accordingly.

3, Molecular features of serrated polyp in colitis affected and unaffected segments were interesting. Molecular features of invasive cancer should be examined.

Response:

Reviewer #2 for the insightful comments, which have helped us improve the results and discussion in our manuscript.

In accordance with your recommendation, we examined the KRAS and BRAF mutations and CIMP-positive status of invasive caner in colitis-affected segments. 

Sixteen invasive carcinoma samples with adequate DNA quality and quantity were examined. KRAS and BRAF mutations and CIMP-positive status were observed in 13% (2/16), 0% (0/16), and 19% (3/16) of invasive carcinomas, respectively, in colitis-affected segments. Considering that the proportions of KRAS and BRAF mutations and CIMP-positive status of invasive cancers tended to be lower than those of serrated polyps in colitis-affected segments, most serrated polyps may not be precursors of colitis-associated invasive carcinoma. These results suggest that colectomy is excessive for serrated polyps in colitis-affected segments. However, considering that serrated polyps are known precursors of CRCs in patients without IBD, serrated polyps in patients with UC should be treated endoscopically, as for those in patients without IBD. 

In our hospital, high-grade dysplasia with a hyperplastic polyp was detected in a 19-year-old man (this case was not included in the present study because the lesion did not meet the diagnostic criteria for SSL, TSA, and SD NOS). This case suggests that some serrated polyps in patients with UC, although rare, have malignant potential. To determine the appropriate management of serrated polyps in patients with UC, further accumulation of cases is desirable.

We added the details of these additional examinations in the methods, results, and discussion. We also created a new supplementary table (S4 Table) explaining the results the of genetic and epigenetic examinations of invasive carcinoma. 

Additionally, we revised the total number and percentage of CIMP-positive serrated polyps in colitis-affected segments in Table 4 (original Table 3) because of a text error in the original manuscript (wrong: 6 (50%) → correct: 5 (38%)).

---

## [Decision Letter · Decision Letter 1]

10 Feb 2023

Serrated polyps in patients with ulcerative colitis: Unique clinicopathological and biologial characteristics

PONE-D-22-31131R1

Dear Dr. Kunisaki,

We’re pleased to inform you that your manuscript has been judged scientifically suitable for publication and will be formally accepted for publication once it meets all outstanding technical requirements.

Kind regards,

Hiromu Suzuki, M.D., Ph.D.

Academic Editor

PLOS ONE

Additional Editor Comments (optional):

Reviewers' comments:

Reviewer's Responses to Questions

**Comments to the Author**

1. If the authors have adequately addressed your comments raised in a previous round of review and you feel that this manuscript is now acceptable for publication, you may indicate that here to bypass the “Comments to the Author” section, enter your conflict of interest statement in the “Confidential to Editor” section, and submit your "Accept" recommendation.

Reviewer #1: (No Response)

Reviewer #2: All comments have been addressed

2. Is the manuscript technically sound, and do the data support the conclusions?

Reviewer #1: Yes

Reviewer #2: Yes

3. Has the statistical analysis been performed appropriately and rigorously? 

Reviewer #1: Yes

Reviewer #2: Yes

4. Have the authors made all data underlying the findings in their manuscript fully available?

Reviewer #1: Yes

Reviewer #2: Yes

5. Is the manuscript presented in an intelligible fashion and written in standard English?

Reviewer #1: Yes

Reviewer #2: Yes

6. Review Comments to the Author

Reviewer #1: The authors have responded appropriately to my concerns, providing additional data.

This reviewer thinks this paper is improved.

Reviewer #2: Clinicopathological and biologial characteristics of serrated polyps in colitis-affected segments is interesting. The revised manuscript is improved and acceptable for publication.

7. PLOS authors have the option to publish the peer review history of their article (what does this mean?). If published, this will include your full peer review and any attached files.

Reviewer #1: **Yes: **Yasushi Sasaki

Reviewer #2: No

---

## [Editor Report · Acceptance letter]

15 Feb 2023

PONE-D-22-31131R1 

Serrated Polyps in Patients with Ulcerative Colitis: Unique Clinicopathological and Biological Characteristics 

Dear Dr. Kunisaki:

I'm pleased to inform you that your manuscript has been deemed suitable for publication in PLOS ONE. Congratulations! Your manuscript is now with our production department. 

Kind regards, 

on behalf of

Dr. Hiromu Suzuki 

Academic Editor

PLOS ONE